# Genomic epidemiology of SARS-CoV-2 in Peru from 2020 to 2024

Benjamin Sobkowiak [1,2], Amy Langdon[1], Pedro E. Romero [3], Gabriel Carrasco-Escobar [4], Diego Villa[4], Renato Cava Miller[4], Víctor Cornejo Villanueva [4], Alejandra Dávila-Barclay [4], Diego Cuicapuza[4], Guillermo Salvatierra[4], Luis González[4], Brenda Ayzanoa[4], Janet Huancachoque[4], Pool Marcos-Carbajal[4,5], Juan Carlos Gómez de la Torre[6], Claudia Barletta[6], Stella M. Chenet[7], Rafael Tapia-Limonchi[7], Jorge Ballón [8], Patrick Fernández [8], Rosario Valderrama[8], Mariana Leguía [9], Christopher Delgado-Ratto [10], Eduardo Gotuzzo[4], Carlos Zamudio [4], Willy Lescano [4], César Cárcamo [4], Verónica Hurtado[11], Priscila Lope-Pari[11], Carlos Padilla-Rojas[11], Víctor Jiménez-Vásquez[11], Oscar Escalante-Maldonado[11], Roger V. Araujo-Castillo [11], César Cabezas[11], Caroline Colijn[1] & Pablo Tsukayama [4,12] ✉

## Abstract

**Background:** Peru recorded one of the world's highest COVID-19 mortality rates, with nearly 4.5 million reported cases and 220,000 deaths by March 2024. Understanding the emergence and spread of SARS-CoV-2 variants in this context is key to informing effective public health responses. This study describes the genomic diversity, transmission dynamics, and geographic spread of SARS-CoV-2 in Peru from 2020 to 2024.
**Methods:** We analyzed nearly 50,000 high-quality public SARS-CoV-2 genome sequences collected nationwide between March 2020 and March 2024. Phylogeographic and mutational analyses were performed to identify variant lineages, trace their origins, and map viral movements within and beyond Peru.
**Results:** We show that Peru's epidemic waves were shaped by the emergence of locally evolved variants, including Lambda (C.37), Gamma (P.1.12), and Omicron (XBB.2.6 and DJ.1) sub-lineages. The city of Lima acted as the primary hub for inter-regional spread, accounting for 47.3% of inferred viral movements to other departments, notably Ancash, Cusco, and Piura. Peru was the source of various lineages that spread internationally, primarily to Chile, the USA, and Europe. Mutational analysis highlighted critical mutations in the spike protein, including L452Q and F490S in Lambda, associated with immune evasion and increased transmissibility.
**Conclusions:** This work demonstrates the capacity of genomic surveillance in Peru to detect and track emerging SARS-CoV-2 variants, providing insights into regional and global transmission dynamics in a high-transmission, middle-income country setting. Sustained, cost-effective genomic monitoring, combined with strengthened bioinformatics and laboratory capacity, is essential for pandemic preparedness in resource-limited settings.

## Plain Language Summary

Peru is one of the countries worst hit by COVID-19, with one of the highest death rates from the disease worldwide. We studied nearly 50,000 viral genomes (the complete set of the virus's genetic instructions) collected across the country between March 2020 and March 2024 to understand how the virus evolved and spread between Peru's 25 administrative regions. We demonstrate that new variants, including Lambda, first emerged in Peru and subsequently spread to other countries. The capital city of Lima was the central hub for the virus's spread to other regions of Peru. We identified changes in the genome that may help it transmit better or evade host defences. Our findings demonstrate how genomic surveillance can help in tracking new variants and inform efforts to control future outbreaks.

[1]Simon Fraser University, British Columbia, Canada. [2]University College London, London, UK. [3]Universidad Nacional Mayor de San Marcos, Lima, Peru. [4]Universidad Peruana Cayetano Heredia, Lima, Peru. [5]Universidad Peruana Unión, Lima, Peru. [6]Sequence Reference Lab, Lima, Peru. [7]Universidad Nacional Toribio Rodríguez de Mendoza, Amazonas, Peru. [8]Universidad Nacional de San Agustín, Arequipa, Peru. [9]Pontificia Universidad Católica del Perú, Lima, Peru. [10]University of Antwerp, Antwerp, Belgium. [11]Instituto Nacional de Salud, Lima, Peru. [12]Wellcome Sanger Institute, Hinxton, UK. ✉e-mail: pablo.tsukayama@upch.pe

Peru has been one of the countries most severely impacted by the COVID-19 pandemic, with 4.5 million recorded cases and 220,000 confirmed deaths as of March 2024, and one of the highest mortality rates globally at approximately 670 deaths per 100,000 inhabitants[1]. Early in the pandemic, Peru implemented strict lockdown measures ahead of many other countries in Latin America. However, the healthcare system, already fragmented and strained, faced severe shortages of ICU beds, oxygen, and mechanical ventilators, worsening the crisis[2]. In addition, the government initially relied on rapid serological tests for diagnosis and promoted hydroxychloroquine, azithromycin, and ivermectin as treatment options, all of which later proved to be ineffective[3–5].

The pandemic had significant social and economic impacts on the country. High poverty and informal employment rates, overcrowded housing conditions, and inadequate access to healthcare and sanitation services limited the population's ability to adhere to strict health measures, amplifying virus transmission[6]. Vaccine rollout delays further exacerbated the crisis; by June 2021, after the country's second epidemic wave, less than 4% of adults were fully vaccinated, and Peru had reached the highest rate of COVID-19-associated deaths worldwide[7,8].

Despite substantial efforts to scale up SARS-CoV-2 genome sequencing, Latin America has contributed less than 3% of sequences on the GISAID database, underscoring a critical gap in the region's genomic infrastructure, trained personnel, and necessary support for large-scale surveillance[9,10]. However, collaborations between government and academic laboratories across the region enabled the monitoring of the virus's evolution and spread[11–13]. Distinct viral lineages, such as Gamma/P.1[14], Zeta/P.2[15], Mu/B.1.621[16], and Lambda/C.37[17,18] were first identified in Brazil, Colombia, and Peru, respectively, and dominated the region in 2020–2021, preventing the Alpha/B.1.1.7 variant from spreading widely, as in most other countries[12]. These local variants were later replaced by the more transmissible Delta/B.1.617.2 in mid-2021 and various Omicron sub-lineages, which have continued to circulate into 2025.

Peru achieved significant milestones in genomic surveillance, sequencing its first SARS-CoV-2 genome in May 2020[19] and identifying the Lambda variant in April 2021[17]. Lambda likely originated in Lima in late 2020[20,21] and displayed unique mutations in the spike protein, including the Δ247-252 deletion, L452Q, and F490S, which enhance ACE2 affinity and reduce antibody neutralization, conferring a competitive advantage over other lineages and potentially increasing reinfection rates[22–25]. By late 2021, Lambda had been reported in 43 countries, with over 10,000 genomes uploaded to GISAID. Other lineages, such as Gamma/P.1.12, Omicron/XBB.2.6, and Omicron/DJ.1 are also likely of Peruvian origin and have contributed to subsequent waves of infection from 2021 to 2024[26].

This study leverages a dataset of nearly 50,000 viral genome sequences from across Peru's 25 departments to reconstruct the initial spread of SARS-CoV-2 from Lima, map the emergence of Lambda and other local variants, and track their subsequent global dissemination. Through this collaborative effort, we characterize the evolution and transmission dynamics of the coronavirus in the understudied Latin American region, highlighting the importance of continued genomic surveillance to prevent future pandemics.

## Methods

### Peruvian COVID-19 epidemiological data collection

The daily registry of COVID-19-positive cases and deaths was obtained from Peru's 'Plataforma Nacional de Datos Abiertos' with data updated as of February 24, 2024[27]. For positive cases, records with missing or invalid values (e.g., unrealistic ages, indeterminate locations) were excluded. Diagnosis dates were then parsed and filtered to include only records from March 2020 onwards. Daily case counts were aggregated for each combination of dates and districts, with missing combinations filled with zeros, assuming no cases were reported. Weekly counts were then computed by summing daily counts over epidemiological weeks (EWs). Death data was similarly accessed, cleaned, and processed. The processed datasets covered weekly counts for each district in Peru from EW 10 of 2020 to EW 03 of 2024. Using 2020 projected district-level population estimates, weekly

incidence and death rates were computed at higher administrative levels (province, departmental, and national). District definitions followed the 2017 census, with newly created districts up to 2020 merged with their corresponding parent districts. In the bivariate scale depicted in Fig. 1b, we employed the Jenks natural breaks optimization method to classify incidence and death rates. We applied a classification approach based on nested average computations to determine the weekly incidence classes for the departments shown in Fig. 1c, bottom right panel. For detailed information on these classification methods, refer to the documentation of the "cartography" package in R[28].

### SARS-CoV-2 genomic sequence data

All SARS-CoV-2 genomic sequences were downloaded from the GISAID database. All Peruvian SARS-CoV-2 sequences deposited in the database with collection dates between 5th March 2020 (the date of the first sequence collection date) and 29th February 2024 were downloaded, along with all global sequences of the Lambda variant C.37, Gamma sub-lineage P.1.12, and Omicron sub-lineages XBB.2.6 and DJ.1. Supplementary Data 1 displays all 548 Pango lineages identified in Peru during the study period, along with sequence counts for each. Only sequences with a complete collection date and high coverage (>90% genome coverage, <5% ambiguities) were included. Sequences were aligned to the Wuhan-Hu-1 reference sequence (GenBank Number MN908947.3) using MAFFT v7.520[29] and filtered with goalign v0.3.5[30], retaining sequences with ≤15% ambiguous sites. The final datasets comprised 49,724 sequences from Peru, 9916 Lambda C.37 sequences, 1205 Gamma P.1.12 sequences, 6704 Omicron XBB.2.6 sequences, and 1726 Omicron DJ.1 sequences. The GISAID IDs of all the sequences included in our analyses are listed in Supplementary Data 2.

### Phylogenetic analysis

Maximum likelihood phylogenies were constructed with IQ-TREE v.2.2.6[31] using the '-m TEST' option to determine the optimal substitution model and 1000 bootstrap replicates. Trees were built separately for all Peruvian sequences and each of the global collections of SARS-CoV-2 sub-lineages described previously. Timed phylogenetic trees were built from the maximum likelihood phylogenies using TreeTime[32] with a coalescent skyline model. The substitution rate was set with an initial prior of $4.1 \times 10^{-3}$ substitutions per site, as inferred from plotting the root-to-tip distance against the collection date in the full Peruvian phylogeny using TempEst[33] (Supplementary Fig. 6). All trees were annotated and plotted using 'ggtree' and 'ggplot2' in R.

### Ancestral sequence reconstruction

We performed a discrete character ancestral state reconstruction for all internal nodes of the timed maximum likelihood phylogenies using 'ace' of the APE package in R. Tips of the phylogeny were labeled by either one of the 25 regions of Peru for the analysis of regional movement in Peru or by the country of isolation for the global movements of SARS-CoV-2 sub-lineages analysis. Alluvial plots were produced using the 'ggalluvial' package in R to illustrate movements between a country or Peruvian region at an internal node with an inferred state to a different state at the tip or between nodes with inferred states.

### Map-based visualizations

We developed geospatial visualizations of inferred viral lineage movements from the previous section using Python and the Cartopy, GeoPandas, Matplotlib, Basemap, and Shapely libraries. The national-level map (Fig. 3) illustrates transitions between departments in Peru using official shapefiles obtained from the Instituto Nacional de Estadística e Informática (INEI). The map was projected using the Plate Carrée projection at a scale of 200 km. Departmental circles were scaled according to the number of inferred outgoing transitions, and directional arrows were drawn between regions with line thickness and transparency representing the number of transitions. Population density was also displayed as a blue choropleth

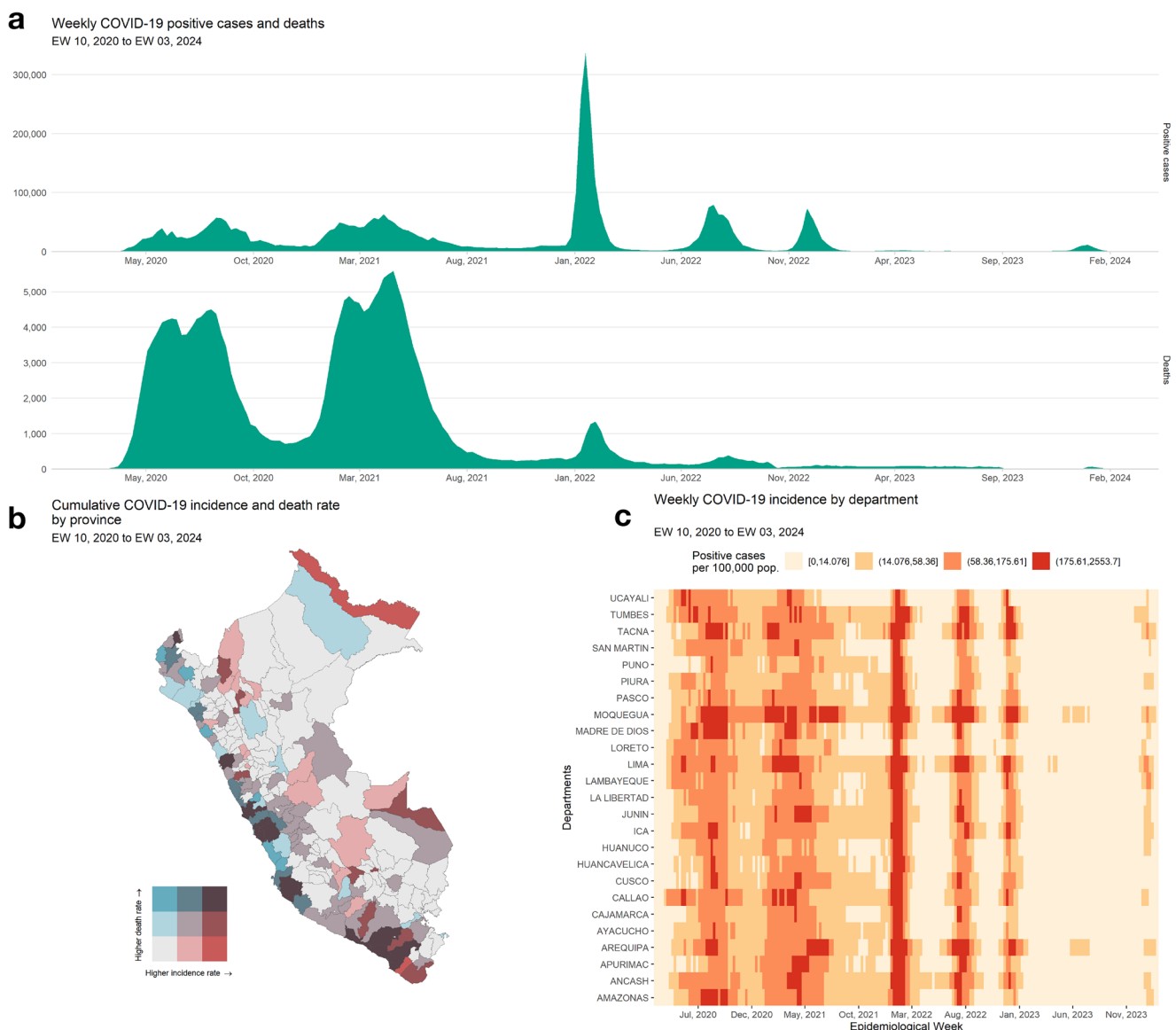

**Fig. 1 | Overview of the SARS-CoV-2 epidemic in Peru between March 2020 and February 2024. a** Weekly COVID-19 positive cases and deaths. **b** Cumulative COVID-19 incidence and death rate by province. **c** Weekly COVID-19 incidence by department.

background using open data from the Ministry of Health of Peru. A logarithmic transformation was applied to better distribute densities across the color range. The international-level map (Fig. 4) displays inferred transitions between countries using geodesic arcs plotted over Natural Earth Data maps, also in Plate Carrée projection and at a scale of 1000 km. The width of each arrow represents the number of inferred transitions, and the color indicates the corresponding viral lineage.

### Sensitivity analysis
To account for potential sampling and sequencing bias in the GISAID dataset, we performed a sensitivity analysis to repeat the ancestral state reconstruction for the regional movement of SARS-CoV-2 in Peru. The maximum likelihood phylogeny was downsampled at the tips by retaining the number of sequences proportional to the population of each region in Peru, as estimated in the 2017 national census (Instituto Nacional de Estadística e Informática, INEI, Peru). This resulted in a total of $N = 8835$ tips being retained. We then reran the ancestral state reconstruction analysis, repeating it 100 times with the retained tips randomly resampled each time before calculating the mean proportion of movements between regions.

### Ethical considerations
This study was conducted in accordance with the ethical principles of the Declaration of Helsinki. The research protocol was approved by the Institutional Review Board of Universidad Peruana Cayetano Heredia (protocol IDs 202151 and 205559, approved in May 2020 and May 2021, respectively). No patient samples or individual-level clinical information were directly analyzed. All analyses were performed exclusively on SARS-CoV-2 genome sequences publicly available through the GISAID platform, accompanied only by non-identifiable metadata (geographic location and collection date). For these publicly accessible datasets, ethics review and the requirement for informed consent were waived by the Institutional Review Board of Universidad Peruana Cayetano Heredia.

## Results
### The COVID-19 pandemic in Peru
The first COVID-19 case in Peru was reported on March 6, 2020, a 25-year-old man living in Lima who had recently traveled to Spain, France, and the Czech Republic. The Peruvian government implemented its first nationwide lockdown on March 16, 2020, which lasted over 100 days. Peru experienced five large waves of COVID-19 infections between 2020 and

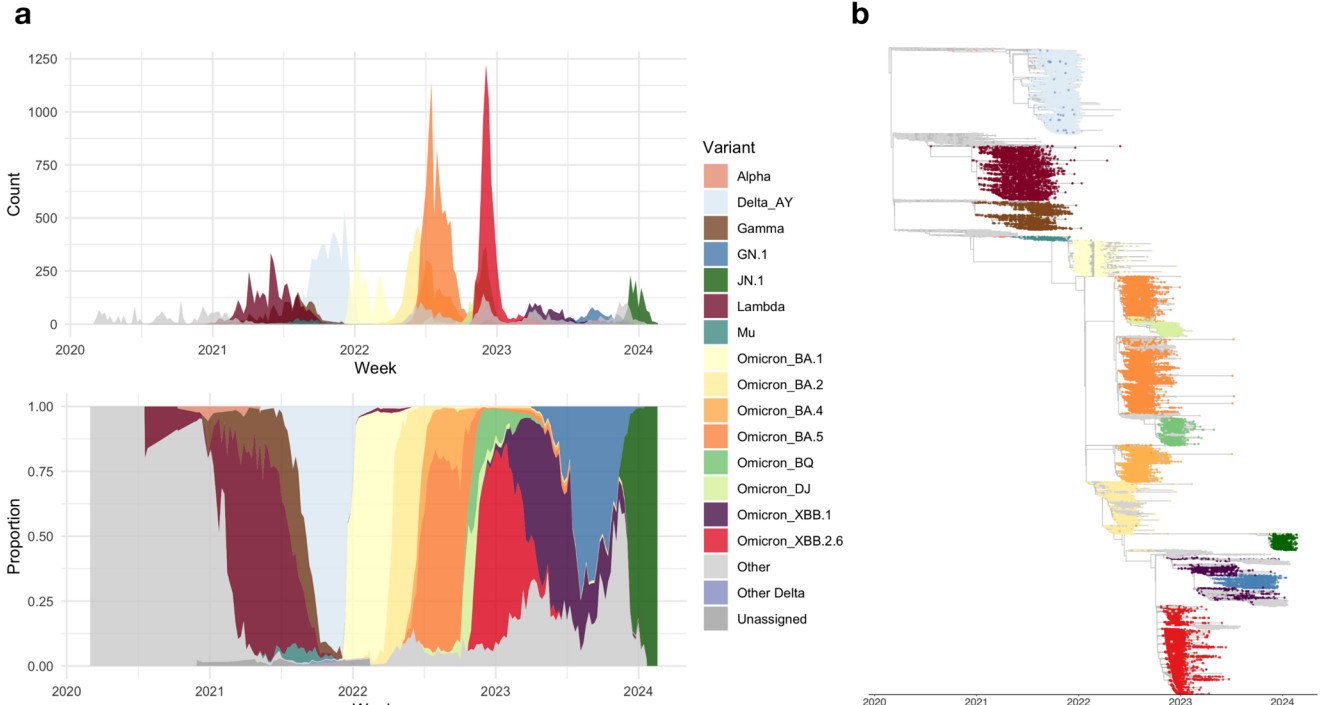

**Fig. 2 | Phylogenetic analysis of 49,724 high-quality SARS-CoV-2 genome sequences collected in Peru between March 2020 and February 2024. a** Weekly counts and proportions of SARS-CoV-2 sequences by variant of concern (VOC) or variant of interest (VOI) collected per week, and (**b**) a timed phylogeny including these sequences, calibrated by collection dates at tips. Tips are colored by VOC/VOI as assigned by Pango lineage calling. Sequences are labeled "Other" for any other lineage that was not a VOC/VOI and "Unassigned" when no Pango lineage could be confidently assigned.

2024, with the third wave being the largest, reaching a peak of 336,436 cases during the third epidemiological week (EW 03) of 2022 (Fig. 1a). Two severe waves of deaths marked the first 2 years of the pandemic, the second of which was the most devastating, peaking at 5595 deaths in EW 16 of 2021. Vaccination campaigns, initiated in February 2021, had a marked impact, as evidenced by the substantial reduction in death counts observed after August 2021. The provinces most affected were located in the coastal departments of Moquegua, Ica, Arequipa, Lima, Ancash, and Tumbes (Fig. 1b). Mariscal Nieto, a province in the Moquegua department, reported the highest cumulative incidence of infections during the study period, with 42,887 cases per 100,000 people. The province of Palpa in the Ica department experienced the highest mortality rate, with 1254 deaths per 100,000 people. Figure 1c shows that the department of Moquegua consistently recorded the highest incidence rates across epidemiological weeks, with a weekly median of approximately 39 cases per 100,000 people and a peak at around 2554 cases per 100,000 people during EW 04 of 2022.

From 2020 to 2024, 49,724 high-quality SARS-CoV-2 genome sequences (i.e., with complete collection date, higher than 90% genome coverage, and fewer than 5% ambiguous base calls) were generated by local public health and university laboratories and deposited in the GISAID database. Figure 2a shows the weekly counts and proportions of sequences collected in Peru, categorized by selected variants of concern (VOC). In total, 548 SARS-CoV-2 Pango lineages and sub-lineages were detected in sequencing data during the study period (Supplementary Data 1).

The first wave of infection was caused by the ancestral Wuhan lineage, which entered Peru multiple times between mid-February and early March 2020, resulting in numerous independent chains of transmission[34]. The second wave, occurring in early to mid-2021, was driven by the Lambda (C.37) and Gamma variants (P.1). The third wave, which began towards the end of 2021, initially featured the Delta variant (AY sublineages) before being rapidly replaced by the Omicron BA.1 variant. Following a brief decline in cases in early 2022, a resurgence of Omicron BA.1 led to a fourth wave, marked by the later emergence of the Omicron BA.4 and BA.5 variants. Case numbers declined towards the end of 2022, followed by a fifth

wave that began in late 2022, dominated by the Omicron XBB.2.6 variant. Through 2023, cases remained relatively low until a small sixth wave emerged at the end of 2023 and into early 2024, driven by the Omicron JN.1 variant. We constructed a time-resolved phylogenetic tree from all Peruvian SARS-CoV-2 sequences to illustrate the evolutionary relationships among variants circulating in Peru during the study period (Fig. 2b). This analysis highlights the dynamic evolution of SARS-CoV-2 in Peru and its distinct waves of variant-driven infections.

## Inter-departmental movements of SARS-CoV-2
To reveal patterns of SARS-CoV-2 transmission in Peru during the pandemic, we analyzed potential movements of infections between the country's 25 administrative regions (departments). We conducted an ancestral state reconstruction on a time-resolved phylogenetic tree to infer the most likely geographic origins (states) at each internal node based on the collection regions for the sequences represented at the tree's tips (Supplementary Fig. 1). This analysis predicted that the earliest SARS-CoV-2 infections originated in Lima, a finding consistent with the first epidemiological reports.

We then inferred movements of SARS-CoV-2 between all departments of Peru for all variants (Fig. 3, Supplementary Fig. 2). These movements indicate cases where the sequence at the tip of the phylogeny was collected from a different region than the inferred region of the preceding node or where the inferred department differs between an internal node and its parent node, suggesting a change in location between infections. Our analysis revealed that most inter-departmental movements originated from Lima, accounting for 41.6% of all cross-regional movements. In contrast, only 25.7% of total movements were from other departments into Lima. Notably, only two departments in the Peruvian Amazon, Loreto and San Martín, showed a higher proportion of movements originating from the region than movements directed into it. Specifically, 4.2% of movements originated in Loreto, compared to 4.1% where Loreto was the destination. Similarly, San Martín accounted for 1.6% of movements originating from the region and 1.5% as the destination. These findings remained consistent

**Fig. 3 | Inferred movements of SARS-CoV-2 infections between the 25 administrative regions of Peru.** The map is projected in Plate Carrée at a scale of 200 km. Circles represent departments and are scaled to the number of inferred outgoing viral movements. Arrows represent the direction and intensity of viral transitions between regions, with width indicating the number of events. Population density is displayed as a blue choropleth background, based on open data from the Ministry of Health of Peru.

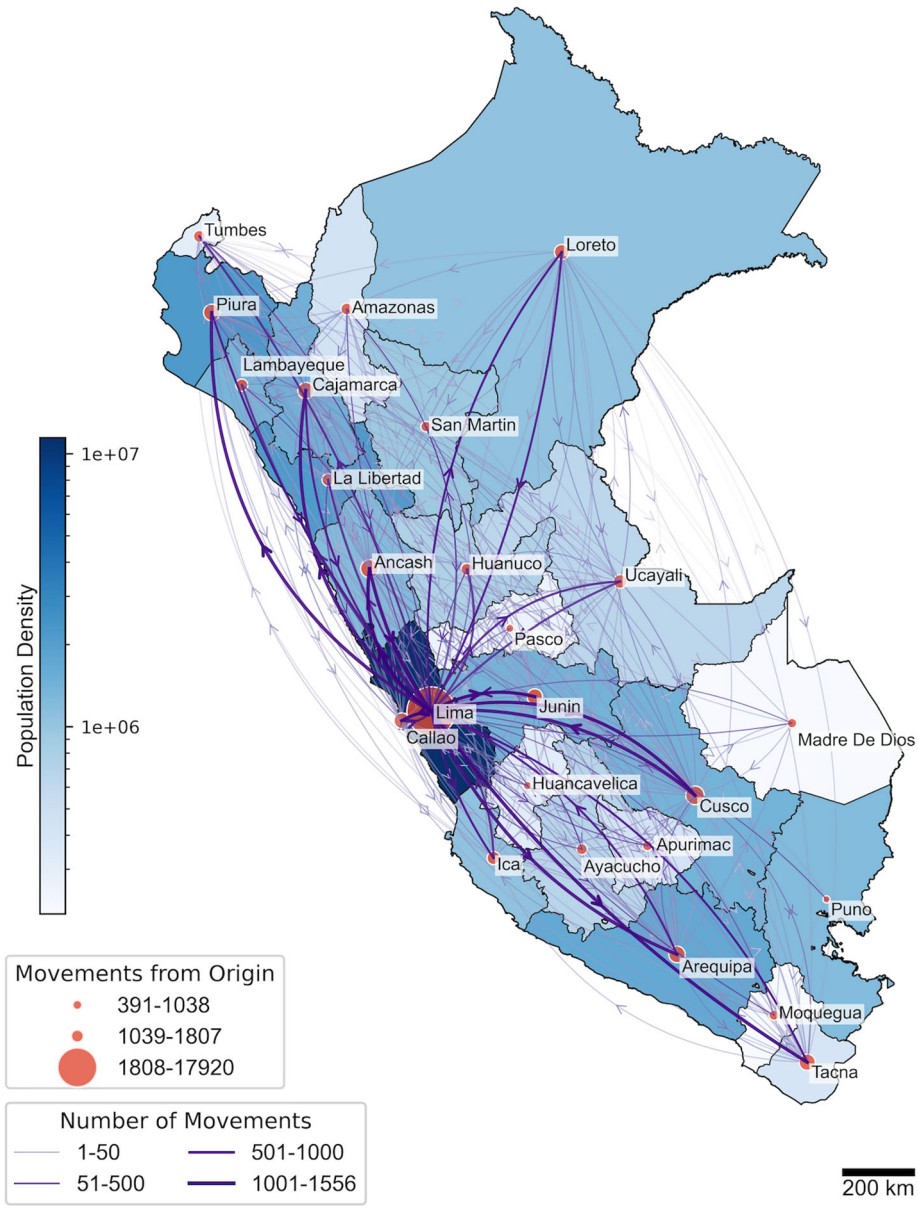

in an analysis focusing solely on inter-departmental movements at the tips of the phylogeny, where the location of sequences was compared only to the inferred location of the preceding node. This approach, as shown in Supplementary Fig. 3, identified Lima as the origin of 47.3% of interdepartmental movements. By analyzing only the tips, we reduced the potential influence of unsampled imported cases from outside Peru, which are more prevalent in deeper branches of the phylogenetic tree.

We found that the largest number of movements originating from Lima were to the adjacent department of Ancash (8.5% of total movements from Lima) and Cusco (8.1% of total movements from Lima), a major tourist destination in Peru. We considered the number of movements that originated from Lima into other departments as a proportion of the total sequences collected in those departments; Puno, Cusco, and Piura all had a high proportion of infections (74.5–76.9%) originating from Lima. These regions all contain sites that are significant destinations for commerce and tourism in Peru.

Phylogeographic analysis using ancestral state reconstruction can be influenced by biases in sampling at the tips of the tree[35]. Thus, we performed a sensitivity analysis to downsample the number of sequences collected in each department (as tips in the tree) relative to the regional population proportion and re-ran the ancestral state reconstruction and inference of inter-regional movements. We found that the results from the full analysis were robust to down-sampling by the population per department. The proportion of inter-regional movements originating from Lima remained high at 44.9% (SD 0.91%), though the proportion of movements with Lima as the destination reduced to 14.9% (SD 0.46%).

## Global movements of viral sub-lineages of Peruvian origin

Next, we examined the origins and inferred international movements of four SARS-CoV-2 sub-lineages likely originating in Peru: Lambda C.37, Gamma P.1.12, Omicron XBB.2.6, and Omicron DJ.1. We performed an ancestral state reconstruction using collections of sequences from GISAID, which included both characterizations as these specific sub-lineages and associated metadata on collection dates and countries of origin. This analysis identified the most likely geographic origins of each sub-lineage and traced their subsequent global dissemination patterns. The ancestral state reconstruction at the root of each time-calibrated phylogenetic tree identified Peru as the most likely country of origin for all four sub-lineages analyzed (Supplementary Fig. 4).

Previous studies have also demonstrated the Peruvian origin of the Lambda variant, which drove a large epidemic wave in early 2021 and later dispersed across South America and 43 countries by August 2021[20,36]. Our

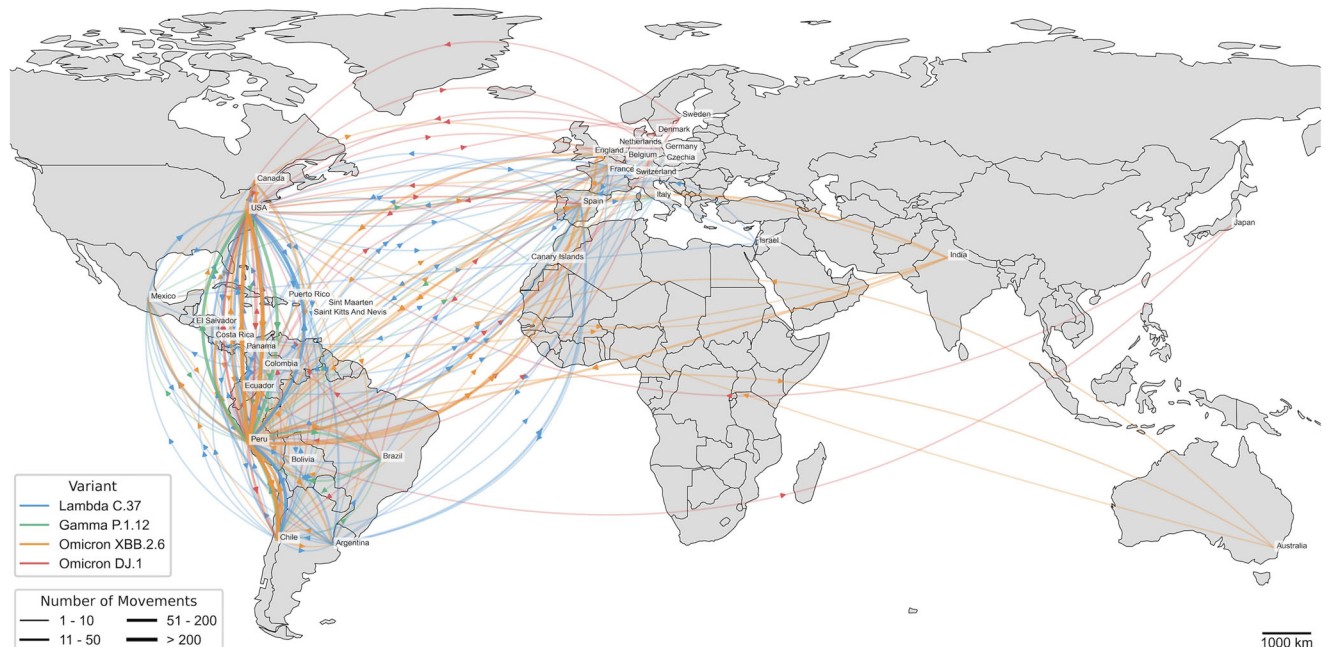

**Fig. 4 | International spread of four SARS-CoV-2 sub-lineages of Peruvian origin.** Movements of Lambda C.37, Gamma P.1.12, Omicron XBB.2.6, and Omicron DJ.1 lineages inferred from phylogeographic analysis. Arcs represent transitions between countries of sample collection, projected in Plate Carrée at a scale of 1000 km. The width of each arrow indicates the number of inferred transitions, and the color denotes the viral lineage.

analysis revealed significant movements of Lambda from Peru to Chile and the USA (Fig. 4, blue lines, Supplementary Fig. 5a). While few transitions were inferred with Peru as the destination, most of these were predicted to originate from Chile, suggesting a large, bidirectional flow of infections across the Peru-Chile border. The Gamma P.1.12 sub-lineage was also found to have originated locally, with the majority of inter-country movements inferred to have originated from Peru (Fig. 4, green lines, Supplementary Fig. S5b). This differs from the ancestral Gamma P.1 lineage, which emerged in Brazil in late 2020[14,36]. Similar to Lambda, most Gamma P.1.12 transitions were from Peru to Chile and the USA, though many movements were also directed toward Brazil. The Omicron XBB.2.6 sub-lineage, which emerged in Peru in mid-2021, also showed significant country transitions from Peru to Chile and the USA (Fig. 4, orange lines, Supplementary Fig. 5c). However, it exhibited greater dissemination than Gamma/P.1.12, with more sequences and destination countries. The Omicron DJ.1 sub-lineage, estimated to have emerged in Peru in mid-2022[26], demonstrated even broader international movements (Fig. 4, red lines, Supplementary Fig. 5d). In addition to significant transitions from Peru to Chile and the USA, DJ.1 also spread to destinations such as Canada and parts of Europe. Unlike other Peruvian-origin sub-lineages, DJ.1 showed more movements from other countries, including Sweden, the USA, and Brazil. This suggests a more complex pattern of onward transmission, with significant international spread following its initial emergence in Peru.

### Mutational analysis of viral sub-lineages of Peruvian origin

We characterized the evolution of locally originated SARS-CoV-2 sub-lineages by identifying mutations (SNPs and indels) with a frequency of >90% in Peruvian strains of the four sub-lineages from the previous analysis and <10% in other Peruvian sub-lineages. Supplementary Data 3 provides details on these high-frequency mutations, including the encoded proteins and corresponding amino acid changes. Table 1 highlights high-frequency Spike protein mutations in Peruvian strains of these four sub-lineages. Spike protein mutations, which are crucial for viral attachment, can enhance transmission and impact vaccine efficacy[37].

We identified 14 SNPs in a high proportion of the Peruvian Lambda C.37 strains, including eight point mutations and a six-codon deletion within the spike protein. While the majority of these mutations have been previously characterized in the Lambda variant[24], two synonymous SNPs in the spike protein, P681P and T723T, were found to be almost fixed in the Peruvian Lambda C.37 strains. In the Gamma P.1.12 variant strains, we identified 23 SNPs unique to this sub-lineage, including nine in the spike protein. These SNPs have been previously described as characteristic mutations of the Gamma P.1.12 variant[14,38]. There were 20 SNPs found to be near fixation in the Peruvian Omicron XBB.2.6 sub-lineage strains, with 11 non-synonymous mutations found in the spike protein. Finally, four mutations were identified as specific to Omicron DJ.1 strains in Peru, including a single non-synonymous SNP in the spike protein, K444N[26].

## Discussion

The COVID-19 pandemic had a profound impact on Peru, resulting in one of the highest per-capita death rates globally, rises in poverty indicators, and significant contractions in GDP[2,7]. This study provides an overview of the transmission and evolution of SARS-CoV-2 in Peru, leveraging nearly 50,000 genome sequences generated and shared between March 2020 and February 2024 by a local network of public health and university laboratories. Our analysis provides insights into the emergence, evolution, and global spread of variants originating from Peru, highlighting significant genomic diversity and inter-regional movements.

The five epidemic waves of COVID-19 in Peru were marked by shifts in variant dominance, reflecting the interplay of viral evolution, public health measures, and population immunity. The ancestral Wuhan strain initially drove high infection and seroprevalence rates, with some of the highest global attack rates reported globally during the first wave in 2020[39,40]. By early 2021, Lambda (C.37) and a local Gamma sublineage (P.1.12) had dominated the second wave, diverging from other regions where Alpha (B.1.1.7) had become predominant[41]. Subsequent waves saw the introduction of Delta (B.1.617.2 and AY sublineages) and Omicron sublineages (BA.1, BA.2, and their descendants) that continue to circulate in 2025. Unlike in most countries, Delta did not lead to a third major wave in Peru, likely due to high immunity from earlier infection waves and increasing vaccine coverage by mid-2021. These shifts in lineages from 2020 to 2024 reflect an evolutionary transition from emergence to endemicity: from variants optimized for transmissibility in a novel host in 2020–2021, such as Alpha, Gamma, and Lambda, to those favoring immune evasion, like the

**Table 1 | Mutations in the SARS-CoV-2 spike protein identified at high frequencies in Peruvian strains of each of the four sub-lineages of Peruvian origin**

| Position in Wuhan-1 reference strain | Sub-lineage | Mutation | Nucleotide in sub-lineage | Proportion of sub-lineage strains carrying nucleotide | Proportion of other strains carrying nucleotide |
|---|---|---|---|---|---|
| 21786* | Lambda C.37 | G75V | T | 0.968 | 0.045 |
| 21789* | Lambda C.37 | T76I | T | 0.987 | 0.043 |
| 22299* | Lambda C.37 | Δ247-252 | - | 0.977 | 0.015 |
| 22917* | Lambda C.37 | L452Q | A | 0.997 | 0.071 |
| 23031* | Lambda C.37 | F490S | C | 0.991 | 0.218 |
| 23403* | Lambda C.37 | D614G | G | 1 | 0.939 |
| 23604 | Lambda C.37 | P681P | C | 0.988 | 0.082 |
| 23731 | Lambda C.37 | T723T | T | 0.999 | 0.045 |
| 24138* | Lambda C.37 | T859N | A | 0.997 | 0.024 |
| 21614 | Gamma P.1.12 | L18F | T | 0.997 | 0.045 |
| 21621 | Gamma P.1.12 | T20N | A | 0.998 | 0.042 |
| 21638 | Gamma P.1.12 | P26S | T | 0.998 | 0.051 |
| 21974 | Gamma P.1.12 | D138Y | T | 0.994 | 0.037 |
| 22132 | Gamma P.1.12 | R190S | T | 0.948 | 0.043 |
| 22812 | Gamma P.1.12 | K417T | C | 0.906 | 0.035 |
| 23012 | Gamma P.1.12 | E484K | A | 0.988 | 0.05 |
| 24642 | Gamma P.1.12 | T1027I | T | 0.999 | 0.037 |
| 25088 | Gamma P.1.12 | V1176F | T | 0.999 | 0.05 |
| 21810 | Omicron XBB.2.6 | V83A | C | 0.995 | 0.076 |
| 22000 | Omicron XBB.2.6 | H146Q | A | 0.999 | 0.087 |
| 22109 | Omicron XBB.2.6 | Q183E | G | 0.947 | 0.064 |
| 22200 | Omicron XBB.2.6 | V213E | A | 0.983 | 0.08 |
| 22320 | Omicron XBB.2.6 | D253G | G | 0.907 | 0.018 |
| 22577 | Omicron XBB.2.6 | G339R | C | 0.989 | 0.075 |
| 22664 | Omicron XBB.2.6 | L368M | A | 0.976 | 0.073 |
| 22895 | Omicron XBB.2.6 | V445L | C | 0.984 | 0.074 |
| 22896 | Omicron XBB.2.6 | V445A | C | 0.983 | 0.083 |
| 22942 | Omicron XBB.2.6 | N460K | G | 0.985 | 0.098 |
| 23019 | Omicron XBB.2.6 | F486S | C | 0.997 | 0.076 |
| 22894 | Omicron DJ.1 | K444N | T | 0.932 | 0.023 |

*Mutations in the spike protein of Lambda that have been previously characterized as lineage-defining.

Omicron sub-lineages, as immunity increased through infections and national vaccination efforts[42–44].

Lima played a central role in disseminating SARS-CoV-2 to other regions, consistent with its status as the capital city, home to one-third of the Peruvian population, and an international transportation hub. This result was robust to potential sequencing bias in Lima, as shown by our sensitivity analysis. Within Peru, we observed major viral movements from Lima to Ancash, Cusco, Callao, Junin, and Piura. These movements were likely influenced by factors such as geographic proximity, population size, tourism, and commercial activity. Similar patterns have been observed in cities such as São Paulo, Buenos Aires, Mexico City, and other major urban centers worldwide, where high connectivity facilitated regional and global dissemination of variants[13,45–49].

Our phylogeographic analysis confirmed the Peruvian origin of four variants: Lambda C.37, Gamma P.1.12, Omicron XBB.2.6, and Omicron DJ.1. Lambda, initially called the 'Andean variant' in the media in 2021, was first identified in Lima in December 2020 and rapidly spread to Chile, Colombia, Ecuador, Argentina, and internationally to the USA and Spain, countries with high air passenger exchange with Peru, indicating that air travel significantly influenced the international spread of variants[50–52]. Key mutations in the spike protein in Lambda, such as the Δ247-252 deletion, L452Q, and F490S, have been shown to enhance immune evasion and transmissibility[24]. Similar adaptive changes were observed in P.1.12 and other Peruvian-origin variants[26], suggesting evolutionary pressures driving the emergence of these lineages toward increased transmissibility and antigenic variation[53]. These findings are consistent with broader regional

trends, where other South American variants, such as Gamma P.1, Zeta P.2, and Mu B.1.621, which also emerged in 2020 in densely populated cities with very high transmission rates, exhibited unique mutational 'constellations' in the S gene and the rest of the coronavirus genome[13,14,36].

Our findings underscore the critical importance of local and sustained genomic surveillance programs for tracking and controlling SARS-CoV-2. Despite major collaborative efforts worldwide, global disparities in sequencing and bioinformatics capacities persist as a significant challenge, with countries such as Peru sequencing far fewer cases than high-income countries[9,10]. To address this, we must develop cost-effective tools that simultaneously track the genomes of multiple respiratory pathogens, such as SARS-CoV-2, influenza, respiratory syncytial virus, and potentially novel viruses, and apply them routinely to human, animal, and environmental samples under a One Health framework. When combined with detailed epidemiological data, genomic surveillance can enhance our preparedness for future epidemics, inform public health interventions, and guide the development of vaccines and therapeutics[54,55].

This study benefits from a robust analytical framework and a large genomic dataset, providing valuable insights into the dynamics of SARS-CoV-2 in Peru. However, a key limitation of our study is the use of maximum likelihood-based phylogenetic methods rather than Bayesian phylodynamic approaches such as BEAST for time calibration and ancestral inference. While Bayesian methods are widely used in viral phylodynamics for smaller datasets, their computational demands scale poorly with data volume. Our dataset includes nearly 50,000 high-quality SARS-CoV-2 sequences, making full Bayesian inference computationally infeasible. Applying such methods would require discarding over 95% of the available data, potentially compromising the spatial and temporal resolution necessary for understanding transmission dynamics across Peru. Our approach, based on maximum likelihood trees and sampling-aware reconstructions, follows current best practices for large-scale genomic surveillance and is consistent with recent national-scale studies[16,56,57].

Additionally, less than 1.3% of the 4.53 million COVID-19 cases reported in Peru have been sequenced, reflecting inherent limitations in genomic surveillance in resource-limited settings. Sampling biases and reliance on publicly available data likely underrepresent the diversity of circulating variants. Future efforts should prioritize improving sampling strategies, linking genomic data with clinical and epidemiological records, and securing resources and personnel to sustain genomic surveillance capacities in the long term. These efforts will be crucial for mitigating the impact of future pandemics and for building more equitable global health security.

## Data availability
Publicly available datasets were analyzed in this study. The number of COVID-19 cases and deaths by region were obtained from Peru's "Plataforma Nacional de Datos Abiertos" (https://www.datosabiertos.gob.pe/group/datos-abiertos-de-covid-19). SARS-CoV-2 genomes used in these analyses were downloaded from the EpiCoV database in GISAID and listed in Supplementary Data 2.

## Code availability
The code used for analyses in this study is available at Figshare: https://doi.org/10.6084/m9.figshare.30615341.v1[58].

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

## Acknowledgements

We thank all the researchers, healthcare workers, laboratory technicians, patients, and individuals who contributed to generating the genomic sequences analyzed in this study. This work was funded by (1) Peru's National Program for Scientific Research and Advanced Studies (PROCIENCIA – CONCYTEC) Contract PE501086419-2024-PROCIENCIA, (2) D43 TW007393 training grant awarded to UPCH by the Fogarty International Center of the U.S. National Institutes of Health, (3) US Centers for Disease Control and Prevention cooperative agreement award GH00266, (4) VLIRUOS JOINT project "Improved infectious diseases control in Peru through sustainable capacity building for bioinformatics and genome sequencing" (PE2019JOI018A102), and (5) Wellcome Sanger Institute International Fellowship awarded to P.T.

## Author contributions

Conceptualization: B.S., A.L., C.C., P.T. Methodology: B.S., A.L., D.V., G.C. Data generation: A.D., D.C., G.S., L.G., B.A., P.M-C., J.H., J.C.G., C.B., S.M.C., R.T., J.B., P.F., R.V., C.D., V.H., P.L., C.P., V.J., O.E. Formal analysis: B.S., A.L., D.V. Visualization: B.S., D.V., G.C., R.C., V.C.V. Funding acquisition: E.G., C.Z., C.D., P.T. Writing (original draft): B.S., C.C., P.T. Writing (review and editing): G.C., M.L., W.L., C.C., E.G., C.Z., P.T. Supervision: G.C., W.L., O.E., R.A., C.C., P.T. All authors read, reviewed, and approved the final manuscript.

## Competing interests

The authors declare no competing interests.
