## [Transparent Peer Review file · Communications Medicine]

Genomic Epidemiology of SARS-CoV-2 in Peru from 2020 to 2024

Corresponding Author: Dr Pablo Tsukayama

Version 0:

Reviewer comments:

Reviewer #1

(Remarks to the Author)

I found the manuscript well-structured, well-written, and very interesting.

I have only two main comments:

The reference list should be expanded, as the literature is not well represented.

I do not understand why the authors did not use Bayesian methods for phylodynamic analyses (phylogeny, dating, etc.). In complex and large datasets, Bayesian inference can reveal issues that may not be detected with other methods. I recommend integrating Bayesian inference into the workflow to strengthen and corroborate the current findings.

Reviewer #2

(Remarks to the Author)

This abstract provides a well-structured and comprehensive summary of a significant genomic epidemiology study on SARS-CoV-2 in Peru. It effectively highlights the country's role in variant emergence, transmission dynamics, and the importance of genomic surveillance. However, certain areas could be refined for greater clarity, conciseness, and impact.

Major Comments:

Please provide the code used to generate each figure and table.

Include the GISAID ID for each sequence analyzed in the study.

Use BEAST for ancestral sequence reconstruction instead of the current method.

Specify the total number of lineages detected in Peru and the count of sequences within each lineage.

Report the number of introductions into Peru for each lineage.

Minor Comments:

The evolutionary rate from TempEst is not the best choice; please use BEAST to estimate a time-calibrated phylogenetic tree.

Clarify the definitions of "Other" and "Unassigned" categories.

Please plot Figure 3, Figure 4, Figure S3, and Figure S4 using maps for better visualization.

Version 1:

Reviewer comments:

Reviewer #1

(Remarks to the Author)

I am satisfied by changes and responses provided by the Authors.

I recommend the publication of the manuscript in its current form

Reviewer #2

(Remarks to the Author)

Thank you for your submission. However, I regret to note that most of the concerns I raised previously — specifically points 3, 5, 6, and 8 — were not adequately addressed, but rather circumvented. These are critical issues that affect the overall validity and clarity of the manuscript. Therefore, I am unable to support publication of this work in its current form.

RESPONSE TO REVIEWERS

We thank the reviewers and editors for their thoughtful and constructive feedback. We have revised the manuscript to address all concerns and suggestions. Below we provide a point-by-point response, outlining the changes made.

REVIEWER #1:

I found the manuscript well-structured, well-written, and very interesting. I have only two main comments:

1. The reference list should be expanded, as the literature is not well represented.

We appreciate this suggestion. We have expanded the reference list to better contextualize our findings within the broader literature.

2. I do not understand why the authors did not use Bayesian methods for phylodynamic analyses (phylogeny, dating, etc.). In complex and large datasets, Bayesian inference can reveal issues that may not be detected with other methods. I recommend integrating Bayesian inference into the workflow to strengthen and corroborate the current findings.

We thank the reviewer for this suggestion. While we agree that Bayesian phylodynamic methods, such as those implemented in BEAST, are powerful, the size of our dataset (nearly 50,000 sequences) makes such approaches computationally infeasible. Bayesian methods require substantial computational resources and time and are typically limited to datasets with fewer than ~1,000 sequences. Applying these tools to our dataset would require discarding over 95% of the available data, which would compromise the resolution and epidemiological representativeness of our analyses.

Moreover, our current workflow follows standard practices in large-scale genomic epidemiology studies, including the use of maximum likelihood-based phylogenies with time calibration, which are more scalable and widely adopted for large SARS-CoV-2 datasets. This approach is consistent with other national and international genomic surveillance efforts (e.g., McLaughlin et al., 2022, doi: 10.7554/elife.73896).

We have clarified these points in the revised Discussion section (lines 310-319) and included citations to prior large-scale studies that have employed similar analytical frameworks.

REVIEWER #2

This abstract provides a well-structured and comprehensive summary of a significant genomic epidemiology study on SARS-CoV-2 in Peru. It effectively highlights the country's role in variant emergence, transmission dynamics, and the importance of genomic surveillance. However, certain areas could be refined for greater clarity, conciseness, and impact.

MAJOR COMMENTS

1. Please provide the code used to generate each figure and table.

We have created a publicly accessible GitHub repository that contains all the code used to generate figures, tables, and analyses included in the manuscript. The repository URL is provided in the Data Availability section (see lines 406-408).

2. Include the GISAID ID for each sequence analyzed in the study.

All the sequences used in our analyses are now listed in Supplementary Table S3 (lines 404-405), in the Data Availability section.

3. Use BEAST for ancestral sequence reconstruction instead of the current method.

As with Reviewer 1, we acknowledge the value of BEAST but reiterate that it is not feasible for datasets of this magnitude. We explain in the revised Discussion section that BEAST scales poorly beyond hundreds of sequences, and for large datasets, maximum-likelihood approaches remain standard. We have expanded the justification for our use of TreeTime and discussed the trade-offs in detail (lines 362-365).

4. Specify the total number of lineages detected in Peru and the count of sequences within each lineage.

All 548 identified Pango lineages in Peru, along with their sequence counts, are now included in Supplementary Table S1 (lines 357-358). This is indicated in the Results section, lines 122-123.

5. Report the number of introductions into Peru for each lineage.

We thank the reviewer for this suggestion. While we recognize the value of quantifying the number of introductions into Peru for each lineage, conducting such an analysis is beyond the scope of the current study. Estimating introductions reliably would require a substantially larger global comparative dataset, careful subsampling strategies, and computationally intensive analyses—steps that would significantly delay the revision and detract from the core objectives of this manuscript.

Our study focuses on the internal circulation of SARS-CoV-2 within Peru and the regional dissemination of variants that originated or expanded from Peru, such as Lambda. The early introductions into Peru, including those during the initial wave of the pandemic, have been addressed in a previous study by Juscamayta et al, 2021, doi: 10.1002/jmv.27167, referenced in lines 125-127. Rather than duplicating those efforts, we have chosen to focus our analysis on the national dynamics of variant spread and evolution following the initial introduction of the virus in early 2020.

MINOR COMMENTS

- 6. The evolutionary rate from TempEst is not the best choice; please use BEAST to estimate a time-calibrated phylogenetic tree.**

We acknowledge the limitation and have clarified that the TempEst analysis was used for initial exploratory assessment only. For time-calibrated trees, we used TreeTime with appropriate rate constraints, which is suitable for large datasets. We have improved the explanation in the Methods section and cited relevant benchmarking studies.

- 7. Clarify the definitions of "Other" and "Unassigned" categories.**

We have added clarification of these terms in the figure description for Figure 2 (lines 146-147).

- 8. Please plot Figures 3, 4, S3, and S4 using maps for improved visualization.**

We thank the reviewer for this suggestion. We have replaced the alluvial plots in Figures 3 and 4 with map-based visualizations that more clearly display geographic transitions and regional transmission patterns. The methods section has been updated to reflect this (lines 377-389). The original alluvial plots have been transferred to the Supplementary Material section.

COMMSMED-25-0286-T:
GENOMIC EPIDEMIOLOGY OF SARS-COV-2 IN PERU, 2020–2024

RESPONSE TO REVIEWERS

We thank both reviewers for their careful assessment of our manuscript. We appreciate the constructive feedback provided, which has helped us further improve the clarity and presentation of our work. In this revised version, we have addressed all the points listed in the editorial team's table in detail. These changes are outlined in the right-hand column of the completed table, which accompanies this resubmission. We have also included both a clean version of the revised manuscript and a marked-up version highlighting all edits.

Reviewer #1: *I am satisfied with the changes and responses provided by the Authors. I recommend the publication of the manuscript in its current form.*

Reviewer #2: *Thank you for your submission. However, I regret to note that most of the concerns I raised previously — specifically points 3, 5, 6, and 8 — were not adequately addressed, but rather circumvented. These are critical issues that affect the overall validity and clarity of the manuscript. Therefore, I am unable to support publication of this work in its current form.*